# A Novel Ginsenoside-Transforming α-L-Rhamnosidase from *Bifidobacterium*: Screening, Characterization and Application

**DOI:** 10.3390/biom14121611

**Published:** 2024-12-16

**Authors:** Chang-Hao Cui, Doohang Shin, Byung-Serk Hurh, Wan-Taek Im

**Affiliations:** 1Sempio Fermentation Research Center, Sempio Foods Company, Osong 28156, Republic of Korea; cchanghao@sempio.com (C.-H.C.); sdoohang@sempio.com (D.S.); 2Department of Biotechnology, Hankyong National University, Anseong 17579, Republic of Korea; 3AceEMzyme Co., Ltd., Academic Industry Cooperation, Anseong 17579, Republic of Korea

**Keywords:** α-L-rhamnosidase, ginsenoside, biotransformation, ginsenoside Rg1, ginsenoside Re

## Abstract

Despite the rapid advancement of glycosidase biotechnology, ginsenoside-transforming rhamnosidases remain underexplored due to a lack of research. In this study, we aimed to bridge this gap by evaluating eight putative rhamnosidases for their ability to transform ginsenosides. Among them, a novel rhamnosidase (C118) from *Bifidobacterium* was identified as being efficient at hydrolyzing ginsenoside Re. This enzyme was expressed well in *Escherichia coli* and exhibited optimal activity at pH of 6.0 and 45 °C. Protein structural predictions revealed that the potential active hydrophobic area near an active pocket may influence the ginsenoside-transforming activities compared to non-active screened rhamnosidases. This enzyme’s thermal stability exceeded that of the only previously known ginsenoside-transforming rhamnosidase, BD890. Additionally, the *k_cat_*/*K_m_* value of C118 was 1.45 times higher than that of BD890. Using recombinant C118 from *E. coli*, all ginsenoside Re in a PPT-type ginsenoside mixture (2.25 mg/mL) was converted after 12 h of reaction. To the best of our knowledge, this is the most efficient ginsenoside Re-transforming α-L-rhamnosidase reported to date, enhancing our understanding of rhamnosidase–substrate interactions and potentially improving the efficiency and specificity of the conversion process. These findings offer promising implications for the production of pharmacologically active ginsenosides in the pharmaceutical, cosmetic, and functional food industries.

## 1. Introduction

Ginseng and its processed products have been utilized as herbal medicine in East Asia for millennia and have gained popularity in Western countries over the past few decades [1,2]. These products are important functional foods, representing more than 50% of the health and functional food market in Korea [3], and are among the most widely consumed herbal nutritional supplements globally [4]. Ginseng and its derivatives exhibit various pharmacological effects, particularly neuroprotective activities, such as enhancing cognition, learning, and memory [5,6]. Ginsenosides, the primary bioactive constituents of ginseng, likely play a key role in their pharmacological activities [6].

Though ginsenoside Re and Rg1 are two major components of these products, comprising similar contents of total ginsenosides in ginseng root [7], Rg1 stands out for its neuroprotective properties, which Re lacks. Rg1 enhances cognitive functions and protects neurons. It has been shown to help Schwann cells recover from oxidative damage induced by hydrogen peroxide via the PKA pathway, potentially aiding nerve regeneration [8]. Prolonged administration of Rg1 notably enhances cognitive function by activating mTOR signaling and increasing the expression of proteins associated with synaptic plasticity in C57BL/6J mice [9]. These findings position Rg1 as a potential therapeutic agent for age-related cognitive decline. Additionally, Rg1 exhibits antidepressant effects in animal models of depression, which are mediated through the activation of brain-derived neurotrophic factor signaling and the promotion of hippocampal neurogenesis [10].

As a result, ginsenoside Rg1 is one of the key marker components in red ginseng and ginseng products according to the Korean Food and Drug Administration (KFDA), which requires the combined content of Rg1, Rg3, and Rb1 to be between 0.8–34 mg/g for red ginseng and Rg1, and Rb1 to be between 2.5–34 mg/g for other ginseng products [11]. While ginsenoside Re is also present in these products with similar contents in ginseng root, it is found in concentrations 3 to 20 times higher than those of Rg1 in ginseng berry extracts [12]. Therefore, converting Re to Rg1 using rhamnosidases could further enhance their functional properties.

Ginsenoside Re, which has an additional rhamnose moiety at the C6 outer position, can be hydrolyzed into Rg1 by rhamnosidases (Figure 1) [13]. Bae et al. primarily reported that a purified rhamnosidase from the *Bacteroides* JY-6 strain hydrolyzed the C-6, α-L-rhamnopyranoside of ginsenoside Re to produce ginsenoside Rg1 [14]. Potential rhamnosidase candidates from three fungal strains for ginsenoside Re transformation have been identified [15]. Subsequently, we cloned and identified a rhamnosidase (Bd890) from *Bifidobacterium dentium* with the ability to convert ginsenoside Re [16]. Bang et al. later found this enzyme’s activity, naming it BdRham from the same species strain [17]. To date, BD890 remains the only known-sequence ginsenoside-transforming rhamnosidase, highlighting the need for further research in this area [18].

α-L-Rhamnosidase (EC 3.2.1.40) is a type of glycoside hydrolase that catalyzes the hydrolysis of terminal non-reducing rhamnoside residues from various natural substrates. They are crucial in various biotechnological applications, particularly in the food, beverage, and pharmaceutical industries [19]. Recent advances in genetic engineering have facilitated the recombinant production of rhamnosidases [20]. Various rhamnosidases have been characterized and applied in the de-rhamnosylation of natural products, such as antibiotics and steroids [21], and in the hydrolysis of naringin in citrus juices [22,23]. They have also been utilized in the production of pharmaceuticals, cosmetics, food products [22], and in the preparation of rhamnose [24].

This study aimed to screen for ginsenoside-transforming rhamnosidases and to deepen our understanding of their mechanisms. Among the eight putative rhamnosidases tested, a novel rhamnosidase (C118) from *Bifidobacterium* was found to exhibit ginsenoside Re conversion activity. This enzyme was characterized and employed for the preparation of ginsenoside Rg1 from a crude PPT-type ginsenoside mixture (PPTGM), a product that can be commercially isolated from crude ginseng extracts. We hypothesize that the use of this enzyme can significantly enhance the efficiency and specificity of the production process of Rg1.

## 2. Materials and Methods

### 2.1. Chemicals and Reagents

The ginsenoside Re (87.6% purity) used as the substrate in this study was obtained from Fusong Biotech Co., Ltd. (Baishan, China). Ginsenoside standards, including Re, Rg1, Rg2, Rh1, F1, and PPT, with a purity of over 98%, were purchased from Nanjing Zelang Medical Technology Co., Ltd. (Nanjing, China). All other chemicals used in this study were analytical reagent grade, with specific sources listed individually.

### 2.2. Bacterial Strains and Vectors

*E. coli* BL21 (DE3) (Enzynomics Co., Ltd, Daejeon, Republic of Korea) was used as the protein expression host, and the pBT7 plasmid (Bioneer Co., Ltd., Daejeon, Republic of Korea) served as the expression vector. A recombinant *E. coli* strain, used for protein expression, was cultured in Luria-Bertani (LB) medium, which is a nutrient-rich solution commonly used for bacterial growth. This medium was supplemented with ampicillin at a concentration of 100 mg/L to selectively maintain the expression of the plasmid and prevent the growth of non-transformed bacterial cells.

### 2.3. Synthesis, Expression and Purification of Candidate Rhamnosidases

Due to differences in codon preference among organisms, the selected eight rhamnosidase genes (Figure 1) were re-encoded into DNA sequences optimized for the codon usage of *E. coli*, thereby enhancing their expression efficiency. These optimized genes were synthesized and cloned into the pBT7 vector, which includes a GST tag, by Bioneer Co., Ltd. (Daejeon, Republic of Korea). Following this, the recombinant vectors were introduced into *E. coli* BL21(DE3) cells, which were subsequently used for protein expression.

The recombinant strains were cultivated in growth medium until the optical density at 600 nm (OD600) reached 0.6, indicating that the bacterial culture had reached the logarithmic phase of growth. At this point, the recombinant protein expression was induced by adding 0.1 mM isopropyl-β-D-thiogalactopyranoside (IPTG), a commonly used inducer that activates the expression of the target gene in the presence of the lac promoter. Additionally, the previously characterized ginsenoside-transforming rhamnosidase, BD890, was also cloned and expressed using the same expression system, for comparison and analysis.

The induced *E. coli* cells were cultured for an additional 18 h at 30 °C to allow sufficient protein expression, after which they were harvested by centrifugation for 15 min at 13,000 rpm. The harvested cells were then washed twice with a 50 mM sodium phosphate buffer (pH 6.0) to remove any residual growth medium. After washing, the cells were resuspended in a solution of 50 mM sodium phosphate buffer (pH 6.0) containing 0.1% Triton X-100 to facilitate cell membrane disruption. The cells were subsequently disrupted by ultrasonication using a Vibra-cell (Sonics & Materials Co., Ltd., Newtown, CT, USA). To remove intact cells and cellular debris, the mixture was centrifuged for 15 min at 13,000 rpm, yielding the crude cell extract for further analysis.

The expressed GST-C118 fusion protein in the cell lysate was isolated using GST·bind agarose resin from Elpis Biotech Co., Ltd. (Seoul, Republic of Korea), which specifically binds to the GST tag for efficient purification. The purity and homogeneity of the protein were assessed by performing SDS-PAGE on a 10% polyacrylamide gel, followed by staining with an EZ-Gel staining solution (Daeillab Co., Ltd., Seoul, Republic of Korea) to visualize the protein bands and confirm the successful purification of the GST-C118 protein.

### 2.4. Prediction of the Active Site and Molecular Docking

The protein structure of C118 was predicted using AlphaFold [25]. Other protein structures were downloaded from UniProt (www.uniprot.org). Subsequently, the residues of the binding site of rhamnosidases were further validated using CB-DOC2 [26]. The docking results are visualized using BIOVIA Discovery Studio v24.1.0.23298.

### 2.5. Influence of pH, Temperature, and Chemical Reagents on C118 Enzyme Activity and Stability

The specific activity of purified C118 was measured using p-nitrophenyl-α-L-rhamnopyranoside (pNPR) as a surrogate substrate. The enzymatic release of p-nitrophenol was monitored using a microplate reader (Bio-Rad model 680; Bio-Rad, Hercules, CA, USA) at a wavelength of 415 nm via a kinetic method. Protein concentrations were quantified using the bicinchoninic acid (BCA) protein assay. All assays were conducted in triplicate to ensure reproducibility.

To assess the effect of pH on enzyme activity, the purified C118 enzyme was incubated with 1.0 mM pNPR as the substrate in 50 mM buffers with varying pH values: glycine–sodium hydroxide (pH 9.0 and 10.0), McIlvaine buffer (pH 8.0), sodium phosphate (pH 6.0, 7.0, and 7.5), sodium acetate (pH 4.0 and 5.0), glycine–HCl (pH 3.0), and KCl–HCl (pH 2.0). The optimal pH for enzyme activity was determined, and the pH stability of C118 was evaluated by incubating the enzyme in each buffer for 1 h at 4 °C and measuring its activity afterward.

Thermostability was assessed by incubating the purified C118 in the buffer at different temperatures for 0.5 h, followed by cooling the sample on ice for 10 min. After this incubation, enzymatic activities were measured using pNPR solution. The influences of temperatures on enzymatic activities were tested by incubating the purified C118 with 1.0 mM pNPR at various temperatures ranging from 4 °C to 65 °C for 5 min at the optimal pH.

### 2.6. Transformation of Ginsenoside Re Using Candidate Rhamnosidases

The crude recombinant enzymes were used to investigate their specificity for hydrolyzing the rhamnose moieties attached to the C6 outer positions of the ginsenosides Rg2 and Re. To assess enzyme activity, enzyme solutions in 50 mM sodium phosphate buffer (pH 7.0) were mixed with an equal volume of ginsenoside solutions (Re and Rg2), each prepared at a concentration of 1.0 mg/mL in the same 50 mM sodium phosphate buffer (pH 7.0). The reaction was carried out at 30 °C. Samples were taken at regular time points (24 h and 48 h) to monitor the progress of the reaction. Each sample was then analyzed by thin-layer chromatography (TLC) after pretreatment steps (such as solvent evaporation or dilution), as outlined in the analytical methods section.

### 2.7. Glycosides Hydrolyzation Preferences of C118

The substrate specificity was evaluated using a variety of chromogenic compounds, specifically 1.0 mM solutions of o-nitrophenyl (oNP) and p-nitrophenyl (pNP) derivatives, as substrates. Reactions were conducted at 45 °C for a duration of 5 min to assess enzyme activity. The substrates tested in this study included an array of different glycoside derivatives, such as pNP-β-D-glucopyranoside, pNP-β-D-galactopyranoside, pNP-β-D-fucopyranoside, pNP-N-acetyl-β-D-glucosaminide, pNP-β-L-arabinopyranoside, pNP-β-D-mannopyranoside, pNP-β-D-xylopyranoside, and pNP-α-D-glucopyranoside. Additionally, several arabinose and rhamnose-based derivatives, including pNP-α-L-arabinofuranoside, pNP-α-L-arabinopyranoside, pNP-α-L-rhamnopyranoside, and pNP-α-D-mannopyranoside, were included. The substrates also covered a range of oNP derivatives, such as oNP-β-D-glucopyranoside, oNP-β-D-galactopyranoside, oNP-β-D-fucopyranoside, and oNP-α-D-galactopyranoside, all sourced from Sigma-Aldrich (St. Louis, MO, USA).

### 2.8. Kinetics of Substrates Hydrolyzation of C118

To determine the catalytic efficiency of C118, the substrate conversion rates (*K_m_* and *k_cat_*) were measured using ginsenoside Re and p-nitrophenyl-α-L-rhamnopyranoside (pNPR) under conditions of pH 6.0 and 45 °C. The enzyme concentration used in these assays was 0.62 mg/mL. Re was tested at concentrations ranging from 0.25 to 10.0 mg/mL, while pNPR was used at concentrations between 0.1 and 5.0 mM. The reaction products, Rg1 and pNP, were quantified, and the *K_m_* and *k_cat_* values were derived by fitting the data to the Michaelis–Menten equation.

### 2.9. Conversion of Ginsenoside Re in PPT-Type Ginsenoside Mixture

The ability of rhamnosidase C118 to convert ginsenoside Re into Rg1 was evaluated using a PPT-type ginseng extract (5.0 mg/mL) containing 2.25 mg/mL of ginsenoside Re as the substrate. The reaction mixture was prepared by adding the cell lysate containing α-rhamnosidase C118 and incubating it at 45 °C. The conversion of ginsenoside Re to Rg1 was monitored, and the conversion rate was calculated by comparing the residual concentration of ginsenoside Re before and after the reaction.

### 2.10. Thin Layer Chromatography Analysis

Thin layer chromatography (TLC) analysis was carried out on 60F254 silica gel plates (Merck, Germany), which are commonly used for the separation of polar compounds. The solvent system used for the mobile phase was a mixture of chloroform (CHCl_3_), methanol (CH_3_OH), and water in a ratio of 65:35:10 (*v*/*v*/*v*), specifically using the lower phase. Samples were applied to the TLC plates by pipetting small aliquots of the reaction mixture onto the plate. To identify the separated compounds, the TLC plates were visualized by spraying with a 10% (*v*ol/*v*ol) sulfuric acid (H_2_SO_4_) solution. After spraying, the plates were heated at 110 °C for 5 min to enhance the visibility of the spots. The positions of the spots were compared to standard ginsenoside samples to identify the components present in the reaction mixture.

### 2.11. High-Performance Liquid Chromatography Analysis

The HPLC analysis was performed using a Prodigy ODS column (150 mm × 4.6 mm, 5 µm). The mobile phases used for the separation were solvent A (acetonitrile) and solvent B (water). A gradient elution program was applied as follows: From 0 to 12 min, 17% solvent A and 83% solvent B; from 12 to 20 min, 25% solvent A and 75% solvent B; from 20 to 30 min, 32% solvent A and 68% solvent B; from 30 to 35 min, 55% solvent A and 45% solvent B; from 35 to 40 min, 60% solvent A and 40% solvent B; from 40 to 45 min, 80% solvent A and 20% solvent B; from 45 to 50 min, 100% solvent A; from 50 to 54 min, holding at 100% solvent A; from 54 to 54.1 min, the composition changed to 17% solvent A and 83% solvent B; from 54.1 to 65 min. The maximum pressure for the HPLC column was set to 200 bars, and the injection volume for each sample was 20 µL. Absorbance was monitored at 203 nm using a UV detector, and the flow rate was maintained at 1.0 mL/min throughout the analysis.

## 3. Results and Discussions

### 3.1. Sequence Identification, Cloning and Activity Comparison

Eight putative rhamnosidases (Figure 2), belonging to the glycoside hydrolase family 78, were synthesized and then inserted into the pBT7 vector for expression. These enzymes were selected based on their homology to the known ginsenoside-transforming rhamnosidase, BD890, with amino acid sequence identities ranging from 45% to 65% (Figure 2). They were recombinantly expressed in *E. coli* BL21(DE3), and the resulting cell lysates were reacted with ginsenoside Re, Rg2 and PPTGM at the concentration of 1.0 mg/mL.

As shown in Appendix A, several rhamnosidases (C118, C172, C185, C188, C440, C770) exhibited rhamnosidase activity against pNPR. However, only C118 demonstrated the ability to transform ginsenosides by hydrolyzing rhamnosides from pNPR, as well as from ginsenosides Re and Rg2. pNPR is commonly used as a substrate for screening rhamnosidases due to its simplicity, speed, and reproducibility, typically assessed through colorimetric assays [27]. Interestingly, in this study, some putative rhamnosidases (e.g., C172, C185, C188, C440, C770) that showed activity against pNPR were unable to hydrolyze other substrates. This suggests that due to substrate differences, the ability to hydrolyze pNPR does not necessarily predict the ability to hydrolyze natural products, and vice versa [15].

### 3.2. Purification and Characterization of Recombinant C118

The rhamnosidase gene of C118, spanning 2667 base pairs, encodes a protein comprising 888 amino acids with a predicted molecular mass of 98.3 kDa and an estimated theoretical isoelectric point (pI) of 5.07, as determined through analysis with the ExPASy tool (https://www.expasy.org/search/tools, accessed on 10 October 2024). The GST-C118 fusion protein was purified through GST·bind agarose resin. After purification, both the supernatant from the cell lysates and the purified enzyme were analyzed by SDS-PAGE to confirm expression and purity (Figure 3). The calculated molecular mass of GST-C118, based on its amino acid sequence, was 123.6 kDa, consistent with the molecular mass observed on the SDS-PAGE analysis. Furthermore, the recombinant GST-C118 accounted for 15.2% of the total soluble protein in the *E. coli* lysate, a relatively high expression level that enhances its feasibility for industrial applications compared to wild-type strains.

The newly identified rhamnosidase, C118, selected from *Bifidobacterium*, could clearly transform Re and Rg2. This transformation was evident from the distinct Rf values observed in the TLC analysis (Figure 4). Specifically, C118 transformed Re and Rg2 into Rg1 and Rh1 by hydrolyzing the attached rhamnose from the rutinose (rhamnose–glucose) attached at the C6 position of aglycons (Figure 5). Additionally, C118 completely converted Re in PPTGM into Rg1 within 48 h. This rhamnosidase belongs to GH family 78 subfamily II according to its amino acid sequence. The products of Re and Rg2 using C118 were also detected using HPLC, and the results are presented in Appendix A, which are consistent with the TLC results. In HPLC analysis, Rg2 and Rh1 cannot be separated from each other as clearly as in TLC.

The optimal pH and temperature for the activity of purified C118 were assessed using pNPR as substrate. The enzyme exhibited activity across a pH range of 6.0 to 7.0 and displayed stability within a slightly broader range of pH 5.0 to 7.0. The highest enzymatic activity was observed at pH 6.0 in sodium phosphate buffer, as illustrated in Figure 6A. Temperature studies revealed that the enzyme’s optimal activity occurred at 50 °C. However, C118 maintained stability only at temperatures below 45 °C, with approximately 94.2% of its activity being lost after incubation at 50 °C for 120 min (Figure 6B). These findings suggest that C118 is a mesophilic enzyme, demonstrating peak performance at moderate pH and temperature conditions, particularly at pH 6.0.

Consistent with observations in other family 78 rhamnosidases from bacterial sources, C118 exhibited a near-neutral optimal pH and a mild optimal temperature [28,29]. At a lower temperature of 20 °C, C118 retained 47.4% of its relative activity. However, its thermostability declined significantly when the temperature exceeded 50 °C, as shown in Figure 6B, similarly to other characterized rhamnosidases within the GH78 family from bacteria [18]. The thermal stability surpasses that of BD890 which is stable below 37 °C. The higher thermal stability of C118 contributes to its superior ginsenoside-transforming activities compared to BD890.

The substrate specificity of C118 was evaluated by testing its activity on pNP- and oNP-glycosides with both α and β configurations. C118 showed no activity towards a range of pNP- and oNP-glycosides, with the exception of pNPR. C118’s inability to hydrolyze other glycosidic bonds from pNP substrates indicates its specificity for certain substrates. This specificity suggests that C118 might be uniquely adapted for targeted biotransformations without affecting other glycosidic compounds, making it a valuable tool for selective processes in biotechnological applications. Therefore, C118 cannot hydrolyze the glucose moieties of ginsenoside Re, which may halt the reaction after hydrolyzing rhamnose, without further hydrolyzing glucose from the aglycon, unlike many naringinases and pectinases [20].

The impact of various chemical reagents on C118 activity was also examined, and the results are presented in Appendix A. C118 showed neither significant inhibition nor enhancement in activity in the presence of SDS, Co^2+^, Na^+^, K^+^, Mg^2+^, Mn^2+^. However, the enzyme activity decreased by 77.8% when 10 mM Cu^2^⁺ was present. Furthermore, the activity of C118 was unaffected by β-mercaptoethanol and EDTA, a well-known inhibitor of thiol groups and a chelating agent, suggesting that sulfhydryl groups and divalent cations are not involved in the enzyme’s catalytic center. As a result, no significant positive effects on C118 activity were observed for the tested metal ions.

Interestingly, the rhamnosidase activities of C118 can be strongly inhibited by Tris ions, with near complete inhibition at 100 mM (Figure 7). Similar results also showed with BD890. Tris-HCl is widely used as a buffer solution component and inhibits a number of enzymes [30,31]. Tris ion inhibition has also been observed in glycoside hydrolases, such as α-galactosidase from *Bifidobacterium longum* subsp. longum [32], sucrase in the intestinal brush border [33], α-galactosidase X [34] and *Bacillus licheniformis* α-amylase [30]. The inhibitory effect of Tris ions is explained by a mechanistic model in which Tris binds at two distinct sites within the active center of glycosidases—specifically, the glucosyl sub-site, which facilitates an interaction between the amino group of Tris and a proton donor in the enzyme’s active center. This binding alters the enzyme’s catalytic function, thereby inhibiting its activity. This is the first report of Tris ion inhibition on a rhamnosidase, highlighting the importance of testing for potential Tris inhibition during the characterization of other rhamnosidases. Such testing could provide valuable insights into the enzyme’s behavior and help optimize conditions for its application.

The kinetic parameters of C118 for substrates pNPR and Re are summarized in Table 1. The *k_cat_*/*K_m_* value offers valuable insight into the efficiency and catalytic mechanism of an enzyme. A higher *k_cat_*/*K_m_* value indicates a more efficient enzyme, capable of processing substrates more rapidly under lower substrate concentrations. The value of *k_cat_*/*K_m_* C118 was found to be 2.45 times of BdRha (also known as BD890; 2.7 min^−1^ mM^−1^), an enzyme previously known for converting ginsenoside Re, which shows the advantage of this enzyme [17].

### 3.3. Ginsenoside Re Transformation Using C118 in PPTGM

Since the solubility of pure Re in water is low (less than 0.2 mg/mL), the reaction was performed with a PPT-type ginsenoside mixture (5.0 mg/mL) containing a soluble 2.25 mg/mL. PPTGM can also be efficiently separated from crude ginseng extracts [35]. As shown in Figure 8, all of the ginsenoside Re was converted after 12 h of reaction at 45 °C by cell lysates of C118, whereas BD890 only converted 54%. This indicates that, as applied in this study, C118 exhibits highly efficient conversion of ginsenoside Re into ginsenoside Rg1.

Ginsenoside Rg1 exhibits more therapeutic activities than Re in various areas, as previously indicated. To achieve Rg1 from Re, numerous researchers have sought to biotransform ginsenoside Re into Rg1 [36]. The bioconversion of Re into Rg1 has been achieved by glycoside hydrolases from various fungal strains [13,37,38,39] and bacteria [14]. However, the lack of efficient enzymes limits the scale-up preparation for the commercial conversion of Re. Compared with the rhamnosidases that have been isolated from original bacteria, the expression level of recombinant rhamnosidase in *E. coli* is significantly higher and offers more benefits for purification and application in the food industry using various genetic tools.

Traditional fungi-derived naringinase and pectinase are well known for their ability to hydrolyze glycosidic bonds in various natural products [18]. However, rhamnosidases like C118 exhibit a higher substrate specificity and efficiency in converting ginsenoside Re to Rg1. This specificity ensures a more targeted bioconversion process, minimizing the hydrolysis of non-target substrates, which is often a limitation with traditional naringinase and pectinase.

### 3.4. Structural Characteristics of Ginsenoside-Transforming Rhamnosidases

The active site was identified by docking the rhamnose into the AlphaFold 3-predicted structure of C118 (Figure 9). When comparing the two known structures of rhamnosidases in the GH78 family, we identified some structural coordinates (Figure 9). Glu478 is likely to act as the catalytic acid in the enzyme’s active site, playing a critical role in the hydrolysis of the glycosidic bond in the rhamnoside substrate.

The single-displacement inverting mechanism proposed for GH78 rhamnosidases involves a key step where a nucleophilic water molecule participates in the hydrolysis of the glycosidic bond. In this mechanism, Asp472 is likely responsible for activating the water molecule, enabling it to attack the electrophilic carbon of the scissile bond. This mechanism is consistent with the typical behavior of glycoside hydrolases in the GH78 family, which often utilize an acid–base catalytic strategy to break down complex sugars. Two aromatic residues, W541 and W600, make extensive hydrophobic interactions with the pyranose ring of the sugar when it binds to L-rhamnose. The pocket topology is completed by additional hydrophobic residues, L597 and W532.

Comparing the predicted protein structures of rhamnosidases in this study, we found that the ginsenoside-transforming rhamnosidases (BD890 and C118) have greater hydrophobicity near the rhamnose-binding pockets (Figure 10) compared to other rhamnosidases, which are either hydrophilic or less hydrophobic than C118 and BD890 (Figure 11). Sequence alignment further revealed a highly conserved hydrophobic cluster (DYV region) region in C118 and BD890, which differs from other enzymes (Appendix A). The mechanisms of ginsenoside transformation by rhamnosidases need further careful studies for confirmations.

## 4. Conclusions

In summary, we report the cloning of a novel ginsenoside-transforming rhamnosidase (C118) from *Bifidobacterium*, classified within glycoside hydrolase family 78. This enzyme was recombinantly expressed and utilized for the biotransformation of the major ginsenoside Re into the pharmacologically active ginsenoside Rg1. The characterization of C118 revealed its optimal reaction conditions to be 45 °C and pH 6.0. Moreover, this enzyme exhibits a high degree of efficiency and specificity in producing ginsenoside Rg1, which subsequently enhances its pharmacological properties. Structural analysis also indicated potential hydrophobic regions near the active pockets, which may interact with the hydrophobic aglycon of ginsenosides, providing insights for improving its activities. By utilizing various emerging immobilization techniques, this enzymatic approach could be effectively applied in the production of ginsenoside Rg1 from relatively abundant PPTGM. This method holds great potential for use in the cosmetics, functional food, and pharmaceutical industries, enabling efficient and sustainable production of ginsenoside Rg1 for a variety of applications.

## 5. Patents

Chang-Hao, C.; Sukchae J.; Doohang, S.; Yunah, K. A composition for converting ginsenosides and method for converting ginsenosides using the same. WO Patent; Application date: 10 July 2024, Sempio Food Company (Seoul, Republic of Korea).

## Figures and Tables

**Figure 1 biomolecules-14-01611-f001:**
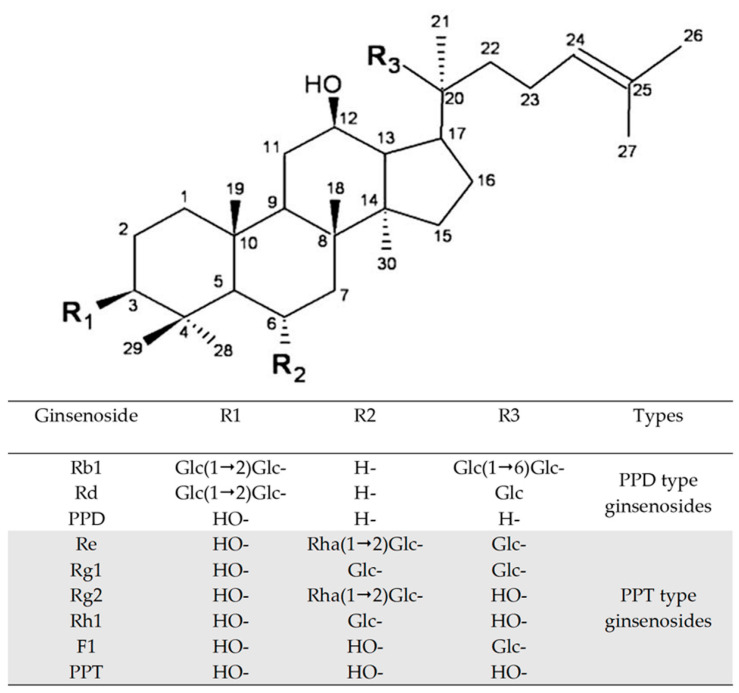
The aglycon and glycon structures of protopanaxadiol (PPD) and protopanaxatriol (PPT) ginsenosides [1]. Glc, β-D-glucopyranosyl; rha, α-L-rhamnopyranosyl.

**Figure 2 biomolecules-14-01611-f002:**
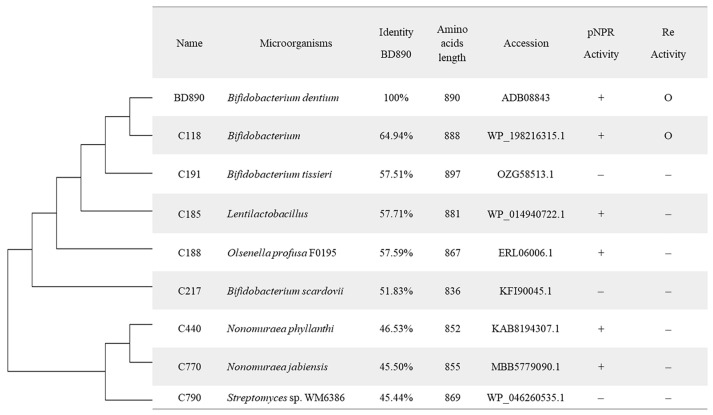
Phylogenetic analysis and activities of eight putative rhamnosidases from glycoside hydrolases in family 78. Amino acid sequences for the study were retrieved from the NCBI and CAZy databases, with their corresponding accession numbers listed in the table. A phylogenetic tree was generated using the neighbor-joining method, employing a Kimura two-parameter distance matrix and a pairwise deletion approach to analyze evolutionary relationships among the sequences. +, indicates the presence of rhamnosidase activity; −, indicates its absence, and O indicates the presence of ginsenoside-converting activity.

**Figure 3 biomolecules-14-01611-f003:**
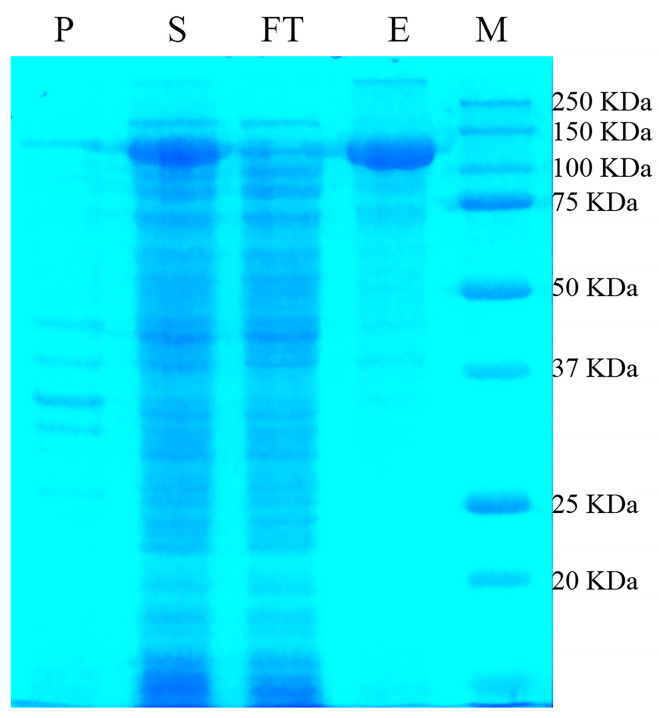
SDS-PAGE analysis was performed to examine the cell extracts containing the recombinant GST-C118 protein and the purified GST-C118. Lane P, the precipitated fraction of the cell extract containing GST-C118; Lane S, the soluble fraction of the cell lysate containing GST-C118; Lane M, protein marker; Lane FT, the flow-through cell lysate of GST-purification resins; Lane E, glutathione elution fractions of GST-C118, respectively.

**Figure 4 biomolecules-14-01611-f004:**
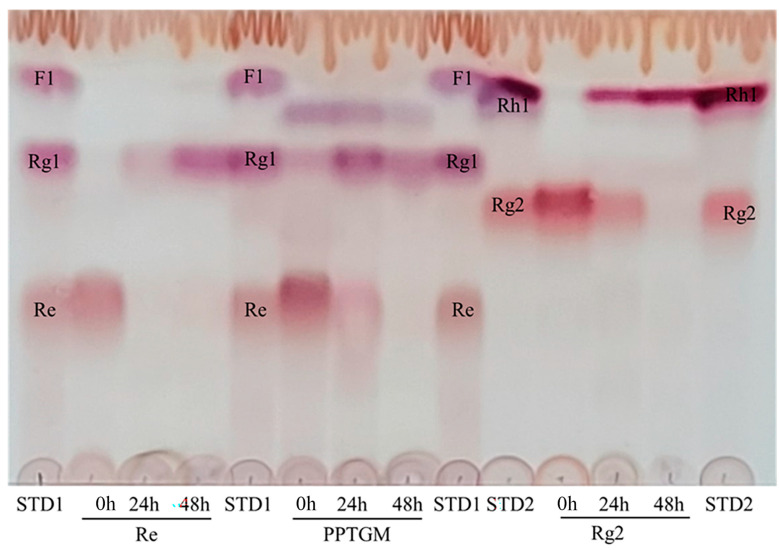
TLC analysis was performed to investigate the biotransformation of ginsenosides Re and Rg2 by recombinant candidates. The reactions were conducted for 24 h using the purified enzyme in 50 mM sodium phosphate buffer (pH 6.0) at 45 °C. The developing solvent system consisted of chloroform, methanol, and water (CHCl_3_-CH_3_OH-H_2_O) in a ratio of 65:35:10. Lanes: Lane STD1, standards of ginsenosides (Re, Rg1 and F1); Lane STD2, ginsenoside standards (Rg2 and Rh1).

**Figure 5 biomolecules-14-01611-f005:**
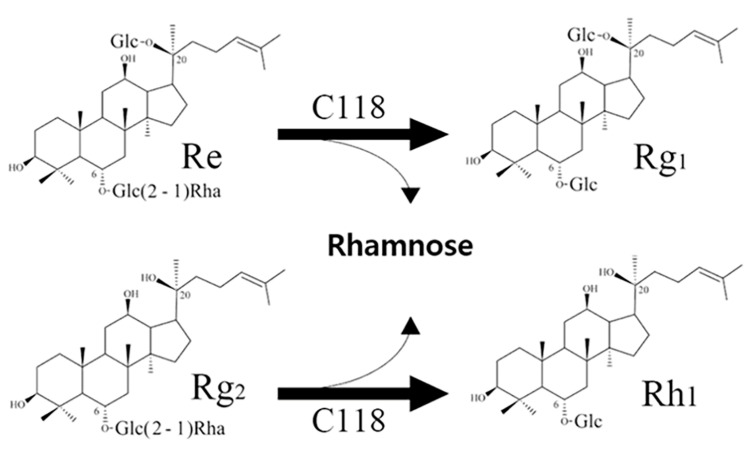
De-rhamnosyl-transformation pathways of Re and Rg2 catalyzed by C118.

**Figure 6 biomolecules-14-01611-f006:**
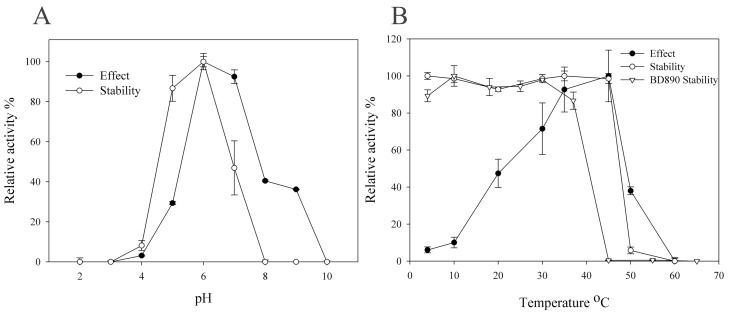
The stability and activity of recombinant C118 were assessed under varying pH (**A**) and temperature (**B**) conditions. Additionally, part (**B**) compares the thermal stability of C118 with that of BD890.

**Figure 7 biomolecules-14-01611-f007:**
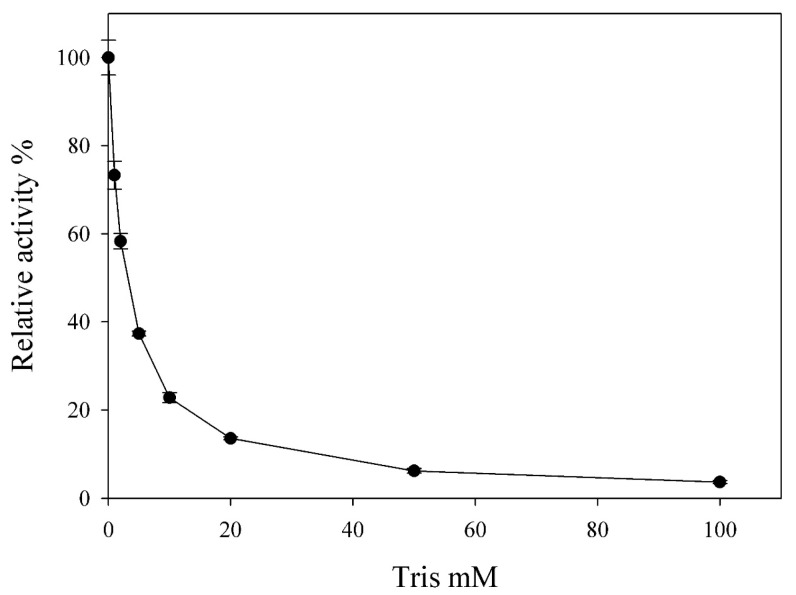
Tris ion inhibition against C118.

**Figure 8 biomolecules-14-01611-f008:**
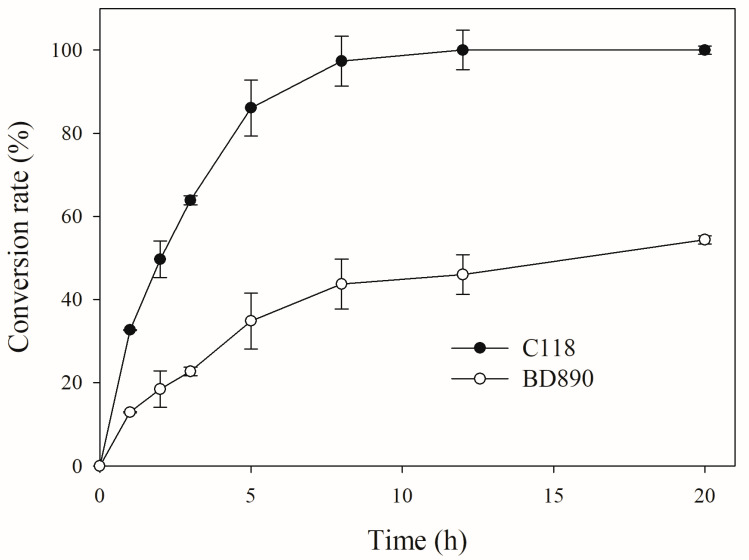
Conversion of Re into Rg1 in a PPT-type ginsenoside mixture using expressed C118 and BD890.

**Figure 9 biomolecules-14-01611-f009:**
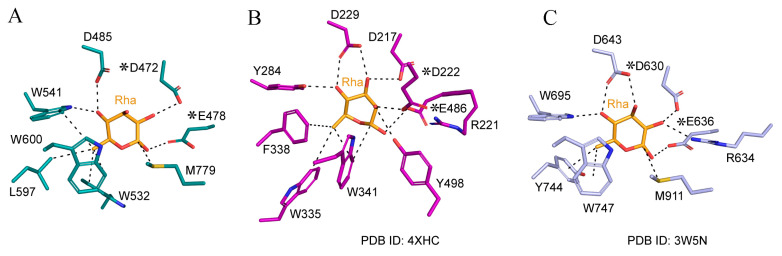
The bound rhamnose structures in the C118-α-L-rhamnose complex (**A**). In panels (**B**,**C**), two GH78 family rhamnosidases, which have been structurally studied with docking with rhamnose, are also presented. The two key residues of rhamnose, based on the binding pose of the rhamnose moiety with C118, are highlighted with (*).

**Figure 10 biomolecules-14-01611-f010:**
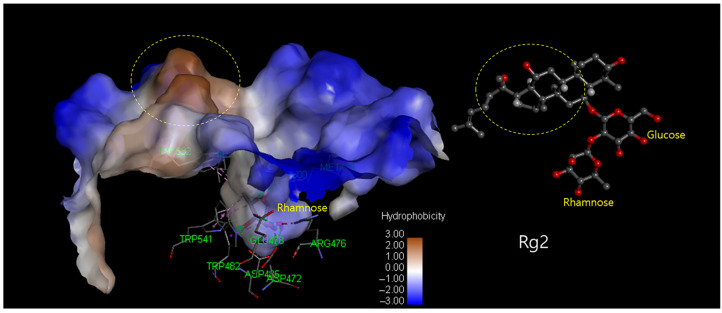
The docking poses of rhamnose located at the active site of C118. Rg2 is presented at the same scale. The hydrophobic region (yellow circle) of C118 is near the hydrophobic aglycon position (yellow circle) when the rhamnose is in the active pocket.

**Figure 11 biomolecules-14-01611-f011:**
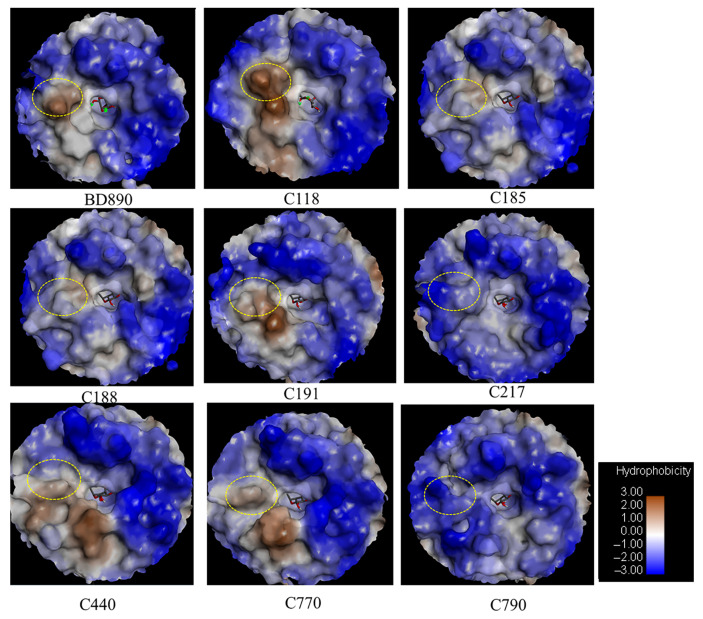
Analysis of the hydrophobic regions of rhamnosidases. The hydrophobic regions of BD890 and C118 are indicated with yellow circles in the same positions.

**Table 1 biomolecules-14-01611-t001:** Kinetic characteristics of C118 against Re and pNPR.

Substrate	*K_m_*	*k_cat_*	*k_cat_*/*K_m_*
	(mM)	(min^−1^)	(min^−1^ mM^−1^)
Ginsenoside Re	0.544	3.61	6.64
pNP-α-L-rhamnopyranoside	2.03	1.24 × 10^6^	0.61 × 10^6^

## Data Availability

All data generated or analyzed during this study are included in this published article.

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
