# Peer review of "A Novel Ginsenoside-Transforming α-L-Rhamnosidase from *Bifidobacterium*: Screening, Characterization and Application"

_biomolecules, 2024, doi:10.3390/biom14121611_

Round 1

Reviewer 1 Report

Comments and Suggestions for Authors

The authors present the identification, heterologous expression and characterization of a novel rhamnosidase capable of ginsenoside hydrolysis. The results are interesting and worth an investifitation. The manuscript should be suitable for publication upon thorough revision. In detail:

1)      The introduction contains several ginsenoside abbreviations and probably many readers will not be familiar with the structural assignment. Therefore, I propose to shift Figure S1 into the main document (as Fig. 1 positioned in the introduction section)

2)      L55: “… it is found in 3 to 20 times higher than Rg1 …” – higher concentrations?

3)      L62: fungi strains – fungal

4)      L63: skip “that”

5)      L64: microorganisms always in italics (also l93, l94, …)

6)      L65: found

7)      L150 / l156: oNP (you may write ortho and para in their first mentioning in l150)

8)      L160 and following: you better use subscript for Vmax, KM and kcat

9)      Results in general: the results section should be structured in a more ordered form and I would propose: 1) Sequence identification, cloning and activity comparison, 2) Purification and characterization of C118, 3) Ginsenoside transformation with C118, 4) Modelling of active sites

10)   L182: The chapter should be called “Results and Discussion”

11)   Chapter 3.1: It would be interesting to know how many similar sequences were found in your database research and why you chose those 8 sequences (your numbering suggests that you fond more putative rhamnosidases?). Are there other sequences more similar to the known BD890?

12)   Modeling: More details should be given about the active site in comparison to known similar structures

13)   L198-201: sentence structure / meaning? – please revise

14)   L207: “… (Figure S4). (Figure S5).” – (Figures S4 and S5)

15)   L233: “… rhamnosidase is belongs …” – skip “is”

16)   Figure 3 and accompagnying text: Nothing is mentioned about the results of the PPTGM transformation (educt composition / product composition)

17)   Figure 3A: What does C mean? Starting value at 0 h?

18)   Figure 3A: Is separation of Rh1 and F1 secure?

19)   Figure 3B: Structures are a bit too small

20)   Table 1 should be skipped – only one compound is active, which is sufficient to mention in the text

21)   Table 2 is “quite boring” and may be shifted into Supplements

22)   L287: tris – better Tris-HCl buffer

23)   L307: consider revising first sentence

24)   Chapter 3.5 is quite short. You should combine this with the initial results outlined in Figure 3 – see proposal for results structuring

25)   Figure 6: Error bars?

26)   From line 340: I would propose to add the headline “Conclusions” here

27)   L426: large space in reference 19

28)   Supplements: Figure S6 on separate page

Author Response

Comments 1. The introduction contains several ginsenoside abbreviations and probably many readers will not be familiar with the structural assignment. Therefore, I propose to shift Figure S1 into the main document (as Fig. 1 positioned in the introduction section)

Response 1: Thank you for highlighting this issue. I agree with your suggestion. Therefore, I have transferred Figure S1 from the supplementary file to the main document and positioned it as Figure 1 in the introduction section.

Comments 2: L55: “… it is found in 3 to 20 times higher than Rg1 …” – higher concentrations?

Response 2: Thank you for pointing out the error. I have corrected the sentence to indicate higher concentrations.

Comments 3: L62: fungi strains – fungal

Response 3: I have corrected it by changing "fungi strains" to "fungal strains."

Comments 4: L63: skip “that”

Response 4: I have corrected it.

Comments 5: L64: microorganisms always in italics (also l93, l94, …)

Response 5: Thank you for your observation. I have reviewed the manuscript and corrected 19 instances where microorganisms should be italicized.

Comments 6: L65: found

Response 6: I have corrected it.

Comments 7: L150 / l156: oNP (you may write ortho and para in their first mentioning in l150)

Response 7: I have corrected the error.

Comments 8: L160 and following: you better use subscript for Vmax, KM and kcat

Response 8: Thank you for your suggestion. I have corrected the manuscript by using subscript for Kₘ, and kₐₜ as recommended.

Comments 9: Results in general: the results section should be structured in a more ordered form and I would propose: 1) Sequence identification, cloning and activity comparison, 2) Purification and characterization of C118, 3) Ginsenoside transformation with C118, 4) Modelling of active sites

Response9: Thank you for your insightful suggestion. I agree that the results section could benefit from a more structured format. Accordingly, I have rearranged the section to follow your proposed structure.

Comments 10: L182: The chapter should be called “Results and Discussion”

Response 10: I have corrected the chapter title to “Results and Discussion” to reflect the content accurately.

Comments 11: Chapter 3.1: It would be interesting to know how many similar sequences were found in your database research and why you chose those 8 sequences (your numbering suggests that you fond more putative rhamnosidases?). Are there other sequences more similar to the known BD890?

Response 11: α-L-rhamnosidases are classified into four glycoside hydrolase families (GHs): GH28, GH78, GH106, and nonclassified (NC), as categorized in the CAZy database (Cantarel et al., 2009; Henrissat and Davies, 1997; http://www.cazy.org/). Among these, the only known ginsenoside-transforming rhamnosidase, BD890, belongs to the GH78 family. Thus, we focused our search for new ginsenoside-transforming enzymes within this family.

The GH78 family contains more than 5000 glycoside hydrolases, with over 100 showing closer similarity to BD890. However, our aim was not just to find highly similar sequences but also to identify enzymes with potentially novel properties. To achieve this, we employed a strategy to screen for enzymes with 40–70% amino acid sequence identity to BD890. This range was chosen to strike a balance: while maintaining sufficient similarity to BD890 for likely functional relevance, it also allowed us to explore sequence variations that could lead to distinct enzymatic characteristics.

From thousands of potential candidates within the GH78 family, we randomly selected eight sequences within this identity range (40–70%). These selections were made to represent diverse sequences with varying degrees of similarity to BD890. Our goal was to evaluate whether these selected enzymes could exhibit ginsenoside-transforming activity and potentially broaden the functional diversity of ginsenoside-transforming rhamnosidases.

By focusing on sequences in this identity range, we aimed to not only identify enzymes similar to BD890 but also uncover new enzymes with unique substrate specificities or other valuable properties. This approach reflects our intent to expand the functional understanding of ginsenoside-transforming enzymes within the GH78 family while avoiding redundancy with already known enzymes.

Comments 12: Modeling: More details should be given about the active site in comparison to known similar structures

Response 12: Thank you for your suggestion. We have identified two studied GH78 rhamnosidase-rhamnose structures. These structures have been included to compare the active site of C118 with them. We have also added a discussion about the results in the manuscript to provide a comprehensive comparison.

Comments 13: L198-201: sentence structure / meaning? – please revise

Response 13: Thank you for pointing out the issue with sentence structure and clarity in the original manuscript. Our goal was to explain that rhamnosidases exhibit diverse substrate specificities, and the ability to hydrolyze one substrate (e.g., pNPR) does not necessarily imply the ability to hydrolyze others (e.g., natural products such as ginsenosides).

In response to your comment, we revised the sentence to improve clarity and better convey this concept. The revised text now explicitly highlights that while some rhamnosidases can hydrolyze pNPR, only C118 demonstrated activity toward ginsenosides Re and Rg2. It also emphasizes the differences in substrate specificities by noting that enzymes capable of hydrolyzing pNPR may not always act on natural products, and vice versa. This revised explanation aligns with the context of our findings and provides a clearer understanding for the reader.

Comments 14: L207: “… (Figure S4). (Figure S5).” – (Figures S4 and S5)

Response 14: I have corrected it by combining the references to Figures S4 and S5 into a single parenthetical phrase: (Figures S4 and S5).

Comments 15: L233: “… rhamnosidase is belongs …” – skip “is”

Response 15: I have corrected the sentence by removing "is" so that it now reads correctly.

Comments 16: Figure 3 and accompagnying text: Nothing is mentioned about the results of the PPTGM transformation (educt composition / product composition)

Response 16: Thank you for highlighting this omission. I have added the necessary information about the results of the PPTGM transformation, including details on the educt composition and product composition, to both Figure 3 and the accompanying text.

Comments 17: Figure 3A: What does C mean? Starting value at 0 h?

Response 17: Thank you for your query. We have clarified the figure by changing “C” to “0h” to indicate the starting value at 0 hours.

Comments 18: Figure 3A: Is separation of Rh1 and F1 secure?

Response 18: Thank you for your observation. In Figure 4, Rh1 and F1 are distinguishable based on their Rf values under the specified TLC conditions (CHCl3-CH3OH-H2O, 65:35:10, lower phase). Although the bands for Rh1 and F1 in the STD1 and STD2 lanes appear close to each other at the start of the TLC, they are clearly separated in the final positions. This separation is evident from the distinct Rf values observed for each compound in the first and last lanes of the standards (STD1 and STD2).Additionally, we performed HPLC in the added data, where you can also find the differences in the HPLC results.

Comments 19: Figure 3B: Structures are a bit too small

Response 19: Thank you for your observation. We have addressed this issue by creating a new figure for Figure 3B, ensuring that the structures are clearly visible and appropriately sized.

Comments 20: Table 1 should be skipped – only one compound is active, which is sufficient to mention in the text

Response 20: We have removed Table 1 from the manuscript and mentioned the active compound directly in the text, as recommended.

Comments 21: Table 2 is “quite boring” and may be shifted into Supplements

Response 21: We have moved Table 2 to the supplementary materials section, as recommended.

Comments 22: L287: tris – better Tris-HCl buffer

Response 22: Due to the restructuring of the “Results and Discussion” section based on comment No. 9, the subtitle has been removed, and some revisions have been made to the sentences. I have also updated the term "tris" to "Tris-HCl " or “tris ions” for clarity.

Comments 23: L307: consider revising first sentence

Response 23: Thank you for your suggestion. I have revised the first sentence in line 307 for clarity and coherence.

Comments 24: Chapter 3.5 is quite short. You should combine this with the initial results outlined in Figure 3 – see proposal for results structuring

Response 24: I have addressed this by combining Chapter 3.5 with the initial results outlined in Figure 3, following the proposed structure for better coherence and clarity.

Comments 25: Figure 6: Error bars?

Response 25: Thank you for highlighting this issue. We have added the missing error bars to Figure 6 to ensure accurate representation of the data.

Comments 26: From line 340: I would propose to add the headline “Conclusions” here

Response 26: Thank you for your suggestion. I have added the headline “Conclusions” at the appropriate place in the manuscript to clearly indicate the conclusion section.

Comments 27: L426: large space in reference 19

Response 27: I have corrected the spacing in reference 19 to ensure it is consistent with the rest of the references.

Comments 28: Supplements: Figure S6 on separate page

Response 28: Thank you for your suggestion. I have moved Figure S6 to a separate page in the supplementary materials section.

Reviewer 2 Report

Comments and Suggestions for Authors

Recommendation: Publish after major revisions noted.

Comments:

In this study, the authors discovered a novel rhamnosidase (C118) from Bifidobacterium, which was efficient at hydrolyzing ginsenoside Re to produce Rg1. They expressed C118 in Escherichia coli successfully and studied its characteristics, and found the enzyme’s thermal stability exceeded that of the only previously known ginsenoside-transforming rhamnosidase BD890, and the kcat/Km value of C118 was 1.45 times higher than that of BD890. This study shows the potential of ginsenoside transformation by C118 in the pharmaceutical, cosmetic, and functional food industries.

The following is specific comments on this paper.

1. It is better to add HPLC spectra for detecting transformation products.

2. Figures S2-S6 are important basis for the paper and should be placed in the text.

3. The Latin used in the paper should be italicized.

4. Molecular formula (CHCl3-CH3OH-H2O and H2SO4) writing should be standardized, numbers should be subscripted.

5. Figure 2 is too large to look coordinated and aesthetically pleasing.

Comments on the Quality of English Language

 The English could be improved to more clearly express the research.

Author Response

Comments 1: It is better to add HPLC spectra for detecting transformation products.

Response 1: Thank you for your suggestion. We have included HPLC data on the transformation products of C118. The HPLC results show that the retention times of ginsenoside Re and Rg1, Rg2, and Rh1 are very close. Therefore, TLC is preferred for detecting the conversions of C118.

Comments 2: Figures S2-S6 are important basis for the paper and should be placed in the text.

Response 2: We have added several figures, including Figures S1, S3, S4, and S5, to the manuscript and revised the content based on reviewer comments. You can find these figures in the revised manuscript. Due to the large number of figures (11 figures), we have kept Figures S2 and S6 in the supplementary file.

Comments 3: The Latin used in the paper should be italicized.

Response 3: We have identified the instances where Latin terms were not italicized through the reviewers' comments and have revised all of them 19 errors accordingly.

Comments 4: Molecular formula (CHCl3-CH3OH-H2O and H2SO4) writing should be standardized, numbers should be subscripted.

Response 4: Thank you for your comment. We have reviewed and revised the molecular formulas in the manuscript to ensure that the numbers are correctly subscripted and that the notation is standardized throughout.

Comments 5: Figure 2 is too large to look coordinated and aesthetically pleasing.

Response 5: Thank you for your observation. We have revised Figure 2 to ensure it looks coordinated and aesthetically pleasing as per your suggestion.

Round 2

Reviewer 1 Report

Comments and Suggestions for Authors

In their revised manuscript the authors answered all issues point by point and improved the quality of the manuscript significantly. I believe that the manuscript is suitable for publication in the current form.

Author Response

Comments 1:

In their revised manuscript the authors answered all issues point by point and improved the quality of the manuscript significantly. I believe that the manuscript is suitable for publication in the current form.
Response 1:

Thank you for your kind and encouraging feedback. We are grateful for your valuable comments and suggestions, which helped us improve the quality of our manuscript. We are pleased to hear that the revised version meets your expectations.

We appreciate the time and effort you dedicated to reviewing our work, and we look forward to seeing it published. Thank you again for your support!

Reviewer 2 Report

Comments and Suggestions for Authors

Accept in the present form.

Author Response

Comments 1:

Accept in the present form.

Response 1:

Thank you very much for your positive feedback and for recommending our manuscript for acceptance in its current form. We sincerely appreciate the time and effort you dedicated to reviewing our work and providing valuable insights throughout the process.

Your thoughtful comments and suggestions have been instrumental in improving the quality and clarity of the manuscript. We are truly grateful for your support and are delighted that the revised version has met your expectations.

Thank you once again for your guidance and encouragement throughout the review process!